# New Therapeutic Targets for Hepatic Fibrosis in the Integrin Family, α8β1 and α11β1, Induced Specifically on Activated Stellate Cells

**DOI:** 10.3390/ijms222312794

**Published:** 2021-11-26

**Authors:** Yasuyuki Yokosaki, Norihisa Nishimichi

**Affiliations:** Integrin-Matrix Biomedical Science, Translational Research Center, Hiroshima University, 1-2-3 Kasumi, Minami-Ku, Hiroshima 734-8551, Japan; teio@hitoshima-u.ac.jp

**Keywords:** fibrosis, integrin, TGFβ, therapeutic target, drug, inhibitor, monoclonal antibody, α8β1, α11β1, hepatic stellate cell

## Abstract

A huge effort has been devoted to developing drugs targeting integrins over 30 years, because of the primary roles of integrins in the cell-matrix milieu. Five αv-containing integrins, in the 24 family members, have been a central target of fibrosis. Currently, a small molecule against αvβ1 is undergoing a clinical trial for NASH-associated fibrosis as a rare agent aiming at fibrogenesis. Latent TGFβ activation, a distinct talent of αv-integrins, has been intriguing as a therapeutic target. None of the αv-integrin inhibitors, however, has been in the clinical market. αv-integrins commonly recognize an Arg-Gly-Asp (RGD) sequence, and thus the pharmacophore of inhibitors for the 5-integrins is based on the same RGD structure. The RGD preference of the integrins, at the same time, dilutes ligand specificity, as the 5-integrins share ligands containing RGD sequence such as fibronectin. With the inherent little specificity in both drugs and targets, “disease specificity” has become less important for the inhibitors than blocking as many αv-integrins. In fact, an almighty inhibitor for αv-integrins, pan-αv, was in a clinical trial. On the contrary, approved integrin inhibitors are all specific to target integrins, which are expressed in a cell-type specific manner: αIIbβ3 on platelets, α4β1, α4β7 and αLβ2 on leukocytes. Herein, “disease specific” integrins would serve as attractive targets. α8β1 and α11β1 are selectively expressed in hepatic stellate cells (HSCs) and distinctively induced upon culture activation. The exceptional specificity to activated HSCs reflects a rather “pathology specific” nature of these new integrins. The monoclonal antibodies against α8β1 and α11β1 in preclinical examinations may illuminate the road to the first medical agents.

## 1. Introduction

Liver fibrosis is an intractable disease with high morbidity by advancing into liver cirrhosis that often causes organ failure, where the parenchymal cells are replaced with collagen species and other matrix proteins. There are currently no approved drugs to treat fibrosis, despite increasing incidence of non-alcoholic steatohepatitis (NASH)-associated liver fibrosis that now causes 2 million deaths per year in the world [1,2,3]. Currently, most drugs in clinical trials target the early steps of steatosis/hepatitis and few target fibrogenesis, itself, especially after simtuzumab (anti-LOXL2) [4] and seronsertive (ASK-1 inhibitor) [5] failed in phase II and III, respectively. In this situation, integrin inhibitors have an emerging therapeutic opportunity in fibrosis [6]. Integrins are receptors for matrix proteins that essentially consist of fibrosis tissues, and some integrins activate latent-TGFβ a central driver of fibrosis [7]. In fact, antagonists for avb1 [8] and avb6 [9,10] showed considerable inhibition in experimental animal models for liver, lung, and kidney fibrosis. Encouraged by the discovery of the TGFβ activation in 1999, pharmacological enthusiasm appears to converge into αv-containing integrins. However, no agents against αv-integrins have yet been approved for fibrosis or other diseases and there are many other integrins that could contribute to fibrosis. Below we revisit the developmental history of αv-inhibitors, and present evidence that two other integrins α8β1 [11] and α11β1 [12], which exhibit pathology-specific expression in fibrosis, merit consideration as targets for anti-fibrotic therapy.

## 2. Leading Three Players in Fibrosis Are Each Related to Integrins

Tissue fibrosis is substantially characterized by deposition of excessive extracellular matrix proteins [13]. The matrix proteins are secreted from activated fibroblasts/myofibroblasts [14]. The fibroblast activation and differentiation to myofibroblast are regulated by TGFβ [15]. Matrix proteins, fibroblasts, and TGFβ, these 3 diverse players in fibrosis, are each functionally dependent on integrins to play their role in development of fibrosis. Integrins are committed as a sensor for the cell-matrix environment [16,17], a signaling receptor of fibroblasts [18,19], and an activator of the latent TGFβ complex [18].

Integrins are a family of cell surface α and β heterodimeric receptors for various extracellular matrix proteins. There are 18 α and 8 β subunits that form 24 heterodimers. The classical role of integrins is cell adhesion through binding to matrix proteins or cell surface immunoglobulin superfamily members, including ICAM-1 and VCAM-1 [19], counter-receptors that hematopoietic integrins use to adhere to vascular endothelial cells [20]. As signaling receptors, integrins mediate fundamental cellular behaviors such as cell migration, proliferation, and survival [21]. There are many matrix proteins which are recognized by multiple integrins. Most integrins can bind multiple matrix proteins and one matrix protein often interacts with multiple integrins. Ligand repertoires of each integrin thus overlap with one another but, of note, the repertoire of most of the 24-integrins is unique. Tissues resident cells, such as epithelial and mesenchymal cells, recognize their matrix environment through integrin receptors and integrins are thus sensors for perturbations in the cellular environment including tissue injury. In healthy tissues, integrins contribute to tissue homeostasis by maintaining tissue integrity, for example, epithelial cells know their position on the basement membrane through integrin signals induced by binding to basement membrane ligands such as laminins and collagen type IV (Figure 1).

Once the basement membrane is injured, however, cells detect damage-associated matrix proteins such as collagen type I and fibronectin. The recognition of tissue injury by integrins also applies to interstitial cells including fibroblasts. Important roles of otherwise resting fibroblasts are to detect perturbations, migrate, and repair tissue injury through secretion of matrix proteins [22]. In these contexts, integrins have long been predicted to play a vital role in the development of fibrosis.

## 3. Activation of Latent TGFβ by Integrins

The discovery that integrins control the activity of TGFβ, the master regulator of fibrosis [23], provided the strongest evidence for contributions of integrins to fibrosis. In 1999, integrin β6 subunit knockout mice (lacking integrin αvβ6 heterodimer) were found to be protected against bleomycin-induced pulmonary fibrosis [24]. These mice were previously found to have exaggerated lung inflammation and so were predicted to have worse fibrosis in response to tissue injury [25]. This seemingly contradictory result was resolved by the discovery of αvβ6 mediated-TGFβ activation. Due to the anti-inflammatory and pro-fibrotic bilateral nature of TGFβ, the lack of TGFβ activation in the β6-knockout leads to both exaggerated inflammation and protection from fibrosis. TGFβ is stored in the matrix milieu encapsulated with pro-domain of the TGFβ gene product (the so-called latency-associated peptide (LAP)) as an inactive homodimer (Figure 2). This manner of storage allows TGFβ to be rapidly activated and act at once on demand without de novo protein synthesis. The mechanisms underlying the TGFβ release from pro-TGFβ was a long-standing controversy, and the discovery of a regulatory system was a big innovation in understanding mechanisms underlying tissue fibrosis.

LAP contains an RGD (Arg-Gly-Asp) tripeptide that αv-containing and other integrins preferentially bind (Figure 2). Binding of cell surface αvβ6 to the RGD in the pro-TGFβ complex, together with contraction of the αvβ6-expressing cell was found to change the conformation of the latent complex, since the other end of the complex is anchored to latent transforming growth factor β binding protein (LTBP) cross-linked to the extracellular matrix [26]. Following release from the complex, TGFβ can bind to its receptors. This is the process called "TGFβ activation". This discovery owes much to a bioassay, where luciferase TGFβ signal reporter cells and β6-transfected cells are co-cultured.

## 4. Trends of Target-Integrins for Fibrosis

An RGD-tripeptide is the first amino acid sequence found as a motif that integrins recognize [27] and is present in many matrix proteins such as fibronectin, vitronectin, and tenascin-C. It is an RGD-peptide that helped the discovery of the first heterodimer by eluting a fibronectin affinity column in 1985 [28]. As many as 8 of 24 members of the integrin family recognize RGD (Figure 3) including all of 5 αv-integrins. This simple linear RGD sequence has played a central role in current thinking about integrin-mediated matrix biology. It seems, therefore, natural that RGD-based pharmacophores have been extensively studied as integrin inhibitors. High hopes for inhibitors of αvβ3 [29] and αvβ5 [16,21,30,31] as anticancer agents due to their proposed anti-angiogenic potential boosted the pharmacological enthusiasm for RGD-based drugs.

Currently, there are three therapeutics venture companies in the US that focus on targeting the integrin family and have developed drugs now in clinical trials [6]. Two of these companies have been well-funded by venture capital and partnerships with pharmaceutical companies. Morphic Therapeutics has a small molecule inhibitor of the α4β7 integrin in clinical trials and Pliant Therapeutics developed a dual inhibitor of αvβ1/αvβ6 for idiopathic pulmonary fibrosis and primary sclerosing cholangitis [32] and a selective inhibitor on αvβ1 for NASH-associated fibrosis. Indalo began a trial of a pan-αv inhibitor that has now been stopped. These concentrated developments on anti-αv-inhibitors are based on many animal experiments and in vitro mechanistic studies, especially αvβ6. A profibrotic role of αvβ6 and an anti-fibrotic effect of the anti-αvβ1 antagonists have been described in lung [9,10], biliary [33], and kidney fibrosis [34,35]. Anti-fibrotic treatment targeting αvβ6 has been effective across organs. However, expression of β6 subunit is restricted to the epithelial cells, not central to fibrosis, and the profibrotic role was found irrelevant to some fibrosis models such as CCl_4_-induced liver fibrosis [33], where fibrosis develops in distance from epithelium. In addition, the pro-TGFβ activation was found to be performed by any of the other αv-integrins, αvβ1, αvβ3, αvβ5 and αvβ8 [36,37,38,39], depending on conditions. To explore further profibrotic effects of the αv-integrins, first, Itgav was deleted selectively in myofibroblasts in pdgfrb-Cre driven conditional αv-knockout mice [39]. The mice lacking all myofibroblast αv-integrins were protected from liver fibrosis. Next, to determine which integrin was important to fibrosis, individual integrins were deleted for αvβ3, αvβ5, αvβ6 (global) and αvβ8 (HSC selective), but no protection was found. Therefore, either a combinatory effect of the αv-integrins or an independent effect of αvβ1 (not deleted due to technical limitation) was hypothesized to drive fibrosis. Shortly, a specific small molecule blockade against αvβ1, C8, was developed and showed an excellent inhibitory effect on murine fibrosis models in multiple organs [8], despite off-target effects of C8 on α4β1 found later [40]. The above history of the RGD peptide starting from the use in the elution buffer, and the established pharmacophore may explain why general interests in integrin-inhibitors converge on αv-integrins. Inhibitors of integrins αvβ1 and αvβ6 are the leading pack in 2021 [6].

## 5. What the History of αv-Integrin Inhibitors Tells Us

αv-integrins activate TGFβ and protect against fibrosis when deleted. Nevertheless, no approved αv-inhibitor for fibrosis and other diseases is in the clinic, despite drugs against integrins, αIIbβ3 [41], α4β1 [42], α4β7 [43], and αLβ2 [44,45] are making big market [6]. It is not easy generally to give specificity to a small molecule inhibitor compared to a monoclonal antibody and it is the case with αv-integrins, with the same pharmacophore shared. The initial αvβ3 inhibitor relied rather on the differential expression in cancer endothelial cells than binding specificity. Furthermore, ligand repertoires of αv-integrins overlap one another, thus signals from αv-integrins could be redundant in some respects. Under these inherent non-specific circumstances of αv-integrins, an interesting idea is to inhibit pan-αv-integrins [46,47]. This reverse thinking appears to reflect the technical difficulty. The pan-αv concept may hold true, because always important is the balance of efficacy and toxicity for drug discovery, depending on the medical need. In fact, IDL-2965 [48] underwent the Phase I clinical trial. In addition, also known for RGD-mimetic integrin inhibitor, is a paradoxical signal input [49,50,51] by binding to the ligand-binding pocket of an integrin. Ligand mimetic inhibitors for αv and other integrins may need to be examined for the unexpected input before the clinical trial.

Among αv-integrins, however, αvβ6 and αvβ8 show a unique preference for ligands, as the binding sequence in pro-TGFβ is RGDLxxL/I [52,53,54]. There are selective inhibitors for αvβ6, a cyclic peptide [55] and a small molecule inhibitor [56], but chemical reagents against αvβ8, so far, can’t be found. Because it hasn’t been long since the therapeutic value of αvβ1 [57,58] was reported, there is only one selective inhibitor published, C8 [8]. C8 retains good selectivity against RGD-recognizing integrins, though it blocks α4β1. As many of pan- and dual- inhibitors for αv-integrins [6,32,59] include αvβ1 in the target, it appears not easy to achieve selectivity to αvβ1. αvβ1 is an expected target as a TGFβ activator integrin on myofibroblasts [8]. PLN-1474, the Pliant’s version of αvβ1 selective inhibitor that completed Phase I clinical trial in March 2021 will tell us whether the selectivity is really important in future clinical trials operated by Novartis. αvβ1 is a kind of atypical integrin in terms of the combination of the subunits, both of which are promiscuous (Figure 3), and most cells store a considerable amount of αv [60] and β1 proteins in the cytoplasm to form various heterodimers. However, αv and β1 do not always form the heterodimer in all cells, by unknown mechanism. Unlike most other integrins, expression of either of the subunits does not identify the heterodimer, and there is no mAb or labeling agent that binds to both subunits at the same time. It is, so far, impossible to define the site of αvβ1 accumulation by immunohistochemistry in the fibrotic and healthy tissues. In contrast, expression of αvβ6 is extremely faithful to epithelial cells [11]. As TGFβ activating ability of αvβ6 is secured, in fibrosis tissues that consist of a lot of epithelial-derived cells, αvβ6 could be a great target as “disease-specific” integrin. Since integrin-mediated TGFβ activation must be crucial for fibrosis, progress of these drugs against αvβ1 and/or αvβ6 in the current clinical trials by Pliant Therapeutics holds a big key to the future direction.

## 6. Disease Specific Integrins

Integrins αvβ6 and αvβ1 are current leading targets for fibrosis. When their antagonists overcome disadvantages from the restricted epithelial expression of αvβ6 and the unclear systemic distribution of αvβ1, the agents would be of great benefit to the public. At the same time, one should be aware that there are 19 non-αv containing integrins. Most of these integrins, excluding leukocyte and platelet integrins, play similar biological roles to αv-integrins in view of cell adhesion, tissue integrity and tissue repair as matrix protein receptors. In addition, three of the integrins, αIIbβ3, α5β1 and α8β1, engage with RGD-containing ligands. It is, therefore, not surprising if some non-αv integrins play roles in fibrosis comparably or more innately than αv-integrins. Is there any subunit that fulfills “disease-specific” expression? Integrin α8β1, unlike most other integrins, shows characteristic restricted expression in mesenchymal cells [61]. A comprehensive gene expression data for 150 primary cells from various tissues [62] reveals α8 subunit is selectively expressed on fibroblasts (Figure 4) [11]. Only α1, α8 and α11 subunits in all α and β subunits of the integrin family show the fibroblast selective expression patterns as in Figure 4. 

Importantly, the α8 expression is, unlike α1, highly upregulated by “culture activation” of rat hepatic stellate cells (HSCs) for 14 days (Figure 5). The marked mRNA upregulation was recapitulated in HSCs from mice and observed at the protein level [11]. HSCs are known to be activated by regular in vitro culture just like they are in fibrotic tissue, associated with elevation of fibrosis markers such as α-smooth muscle actin (α-SMA). The ”pathology-specific” induction of α8β1 suggests the ”pathology-specific” functional property of α8β1. Interestingly, α11 is upregulated similarly to α8 by the same culture activation in rat HSCs [63], with a little bit earlier response than α8. α11β1 is one of the 4 collagen receptor integrins and, of note, selectively expressed in fibroblasts, such as α8β1 (Figure 5). Because collagens are predominant matrix proteins in fibrosis tissues, collagen receptor integrins have been assumed to play roles in fibrosis. We will refer to data by us and others for these 2 integrins to evaluate their suitability as the therapeutic targets. It appears why these 2 integrins have been unattended is not because of their functional limitation but substantially by the absence of specific inhibitors.

## 7. Pathology Specific Integrin α8β1 with TGFβ-Activating Potential

Three non-αv integrins, αIIbβ3, α5β1 and α8β1, recognize RGD sequence. αIIbβ3 (GPIIb/IIIa) is exclusively expressed on platelets. α5β1 interacts with a narrow spectrum of ligands and specifically recognize the RGDS sequence in the 10th type III repeat of fibronectin and does not recognize RGDL in TGFβ pro-protein. Integrin α8β1 more promiscuously engages with RGD in many proteins including, nephronectin, fibronectin, osteopontin, tenascin-C, Mfge-8, and, of note pro-TGFβ, protein.

### 7.1. Proposed Contribution of α8β1 to Fibrosis and Opposing Findings

In 2000, α8 was first reported for high upregulation at the sites of fibrosis in lung and liver fibrosis [64]. Similarly, α8 induction was observed in activated fibroblasts in tissues of cardiac fibrosis [65] vascular stenosis [66], gingival overgrowth [67]. Due to the prominent upregulation in the tissues in activated fibroblasts/HSCs, integrin α8β1 was expected to be a new therapeutic target of fibrosis [67]. However, the expectation was opposed by two findings. First, α8β1 was reported not to activate TGFβ. A bioassay using SW480 α8-transfected colon cancer cell line showed negative results unlike αvβ6, despite α8β1’s recognition of RGD in pro-TGFβ [68]. Second, the expected reduction in fibrosis was not observed in a global Itga8-null mice line in heart [69] and kidney [70] fibrosis.

By our recent experiments, however, α8β1 activates TGFβ, in a cell type-specific manner [11], where α8β1 on fibroblasts/HSCs activates TGFβ in contrast to no activation by α8-transfected SW480 as reported [68]. Cell contractility is a critical force for integrin-mediated TGFβ activation as demonstrated between αvβ6 and pro-TGFβ protein in a 3D model based on crystal structures [54]. Interestingly, however, αvβ6 expressed on non-contractile SW480 cells activates TGFβ, which is disrupted by cytochalasin D [11]. The TGFβ activations in different manner by α8β1 from αvβ6 should be explored for the better understanding of the integrin-mediated TGFβ activation. The other set of conflicting results are from the Itga8-null mouse line. This could be attributed simply to differential effects of α8β1 by organs. However, a special phenotype of the Itga8 knockout mouse line [71] used in those experiments needs to be assessed carefully. The authentic knockout mice line was established in 1997 by crossing Itga8^+/-^ heterozygous mice, which totally lacked α8 expression after the time of the fertilization. Interestingly, the line is known for bilateral fatal kidney agenesis, suggesting a role of α8β1 in nephrogenesis. This effect is corroborated by the discovery of recessive mutations of ITGA8 in families with kidney agenesis [72]. Of note, the bilateral agenesis in the KO line occurs only in about half of the mice at birth and the other half are survived with one or two kidney(s). The survived latter half mice were naturally used in the experiments. As the number of kidneys indicates, the effects of Itga8-deletion could be different between the fatal and survived groups at least on nephrogenesis. Besides the kidney agenesis, the effect of the deletion in the survived mice was likely to be compensated by ”a stochastic factor” [71]. Such compensation for tuning in the molecular network is commonly found in a genetic model of zebrafish, which was, notably, less observed in the siRNA knockdown model [73]. The zebrafish compensation only in the genetic model supports the idea that the no fibrosis attenuation in the Itga8^−/−^ mice line was biased by compensation in the molecular network. In addition, the surviving Itga8^−/−^ mice are fertile and were maintained under mutual Itga8^−/−^ mating [74]. Since the mice were not congenic, the maintenance could have concentrated the genetic background of the founders related to the stochastic factor. In our Tamoxifen-inducible Itga8^flox/flox^; Rosa26-Cre mice, α8 expression is preserved until the beginning of the experiment, kidney development is normal, and importantly liver fibrosis is attenuated [11].

### 7.2. Neutralizing mAb for α8β1

Most integrin heterodimer receptors have been characterized for their function by the use of a specific neutralizing monoclonal antibody (mAb), which is generally obtained following molecular cloning of a subunit and identification of the heterodimer [75]. However, no one has successfully generated the neutralizing mAb against α8β1. Therefore, although literature described α8 expression in activated fibroblasts, there was no functional evidence for the profibrotic role of α8β1. We, therefore, immunized avian species, chicken, with murine α8 protein and screened with human α8β1, consequently obtained 3 neutralizing clones [76]. The epitope mapping revealed that the epitopes of 3 independent mAb clones were partially overlapped. And the aminoacid sequences at the top of the extended loop of the α8 β-propeller domain were totally the same across mammalian species. The conserved sequence in the epitope explains why preceded efforts of others in mice, rats, and rabbits were unsuccessful. The mAb clones, YZ3, YZ5, and YZ26 show potent blocking activity of IC_50_ < 0.1 μg/mL for cell adhesion, indicating sufficient potency in vitro and in vivo experiments.

### 7.3. A Role of α8β1 in Fibrosis

Profibrotic roles of α8β1 were found in vivo and in vitro with the mAb, YZ3 [11]. First, we injected the mAb in 3 liver fibrosis models, biliary duct ligation (BDL), CCl_4_ and clinically relevant NASH-associated model. Liver fibrosis in each model was attenuated by 10 mg/kg injection twice a week. Expression of α8 in clinical liver fibrosis was analyzed in 90 patients who had undergone hepatectomy and elevated in the fibrotic livers compared to F0 controls. Several reports indicate α8 expression is associated with a contractile phenotype of cells such as arrector pili [77] and sensory hair cells [78]. We, therefore, evaluated the contribution of α8β1 to myofibroblast differentiation in HSC culture activation. In three markers upregulated, Acta2, Col1A1 and extra domain-A of fibronectin (EDA), we found Acta2 was specifically reduced by the anti-α8 mAb. To ensure contribution of α8β1-mediated signal input, we plated HSCs on an α8β1 ligand, nephronectin. This combination of interaction induces nephrogenesis and is biologically active [79]. As expected, the HSCs induced Acta2 expression dose-dependently on nephronectin, which was abrogated by YZ3. The α8β1-induced myofibroblast differentiation was confirmed by gel contraction assay, where gel contraction induced by nephronectin in collaboration with TGFβ was inhibited by YZ3. Taken the myofibroblast differentiation potential with the TGFβ activation as described above together, α8β1, at least in part, drives liver fibrosis. Consistent with our finding, recently, two independent groups reported the profibrotic role of α8β1 in liver fibrosis [80,81].

## 8. Pathology Specific Integrin α11β1 with a Property of Collagen Receptor

In all the 24 integrins, α11β1 and α8β1 are the only members that are restricted within the mesenchymal tissue. And they are selectively expressed in fibroblasts. Interestingly, moreover, both are the only integrins that are highly induced by culture activation in HSCs (Figure 5) [11]. The ligand repertoire of α11β1 is, however, distinct from α8β1. α11β1 is one of four collagen receptor integrins. Collagen types I and III are representative matrix proteins that deposit excessively in fibrosis. Changes in the collagen density of fibrotic tissue are sensed at least in part by the receptor integrins. There are 4 integrins that exclusively serve as collagen receptors [82], α1β1, α2β1, α10β1 and α11β1 [83]. We [11] and others [63] found, unlike α11β1, the other 3 integrins are not highly induced during culture activation of HSCs (Figure 5). Tissue distribution of α1 is relatively selective in the mesenchyme but expressed also in other cell types such as neurons, leukocytes, and endothelial cells (Figure 4). α2 is expressed ubiquitously in epithelial cells, endothelial cells and fibroblasts and mesenchymal stem cells. α10β1 is a specific receptor for collagen type II that is a predominant constituent of the cartilage tissue (<50%) and expressed largely in chondrocytes. α11 is known for its mesenchymal-specific expression [84]. Compared to α8, α11 is expressed more diversely in mesenchyme including chondrocyte, smooth muscle cells, adipocytes, and mesenchymal stem cells (Figure 4).

The character of α11β1, a receptor for collagen at least types I, III, and V [83,85] and highly induced in fibrosis, strongly suggests that α11 could modulate development of fibrosis. There is accumulating evidence showing the profibrotic property of α11β1; for example, α11β1 promotes myofibroblast differentiation [86,87] and the expression is regulated by TGFβ [88,89]. Furthermore, α11 accumulates in fibroblasts/myofibroblasts in the sites of fibrosis, in rodent models of the liver, lung and kidney [12] and in human gingival overgrowth tissue [67]. In addition, cardiac fibrosis is induced by over-expression of α11 in mice [90]. It is, therefore, not surprising if α11β1 promotes fibrosis. As the last piece of evidence, direct inhibition of in vivo fibrosis by a specific antagonist for α11 would be a definitive value. Alternatively, Itga11 knockout mice with inducible ablation could also serve as the remaining piece. One or both ways of the target validation have to be done to begin the development of the clinical agents targeting α11β1. We have generated a neutralizing mAb against α11β1, clone YW33. The mAb specifically recognizes α11 among the 4-collagen receptor integrins and cross-reacts across mammalian species, at least human, mouse and rat. The mAb inhibits cell adhesion of human α11-transfected C2C12 cells to collagen type I and type III. Interestingly, the anti-α11 mAb detaches cells already adhered from the plates coated with collagen type I and Type III (Figure 6), showing that the mode of action of the inhibition is allosteric. An allosteric inhibition is proposed as an essential requirement for integrin inhibitors [91,92] as described later.

## 9. Future Directions for Anti-Fibrotic Integrin Inhibitor Drugs

Although many anti-fibrotic drugs targeting integrins and other molecules passed through preclinical examinations, there is no drug in the clinic. Dogmas in the preclinical examinations are revisited.

### 9.1. Evaluation in Animal Models

The severity of fibrosis in an animal experiment is evaluated relying largely on collagen deposition in the fibrotic tissues. Several quantification methods of the collagen deposition are established, such as a measurement for hydroxyproline content in the tissue [93] and for areas positive for Sirius red or Masson’s trichrome staining in histology sections. The amount of collagen is the gold standard of the evaluation, as it is the predominant physical constituent of fibrosis. However, because results from preclinical studies do not assure the consequences in clinical trials at all, one might doubt whether the gold standard for the severity holds good as the endpoint for drug efficacy in human fibrosis.

Collagen type I is the essential and final product of the fibrogenic pathway, which inversely indicates that the production is preceded by changes in and around collagen-producing cells to activate the collagen production network. Most clinical fibrosis, including idiopathic pulmonary fibrosis and liver cirrhosis, progresses slowly and steadily [94], sometimes over decades. In contrast, animal models are set to develop the pathology in several weeks treated with such as toxic chemicals or by surgical cholestasis. Considering the chronicity in humans, the rapid and powerful activation of the collagen-producing network in the animal models could be regulated by a distinct mechanism from human liver fibrosis/cirrhosis. With this regard, should the efficacy of an anti-fibrotic drug be judged by the amount of collagen deposition in the animals? Why is an ability to suppress chronic collagen deposition in human reflected in the amount of collagen deposition in the animal model? What we ought to look into animal experiments is not the consequence after “weeks” but changes in the network and its mechanism that are shared in human. Some existing makers might reflect the efficacy better and predict more of the clinical trials. Minor changes in gene expression are detectable, besides comprehensively, which was impossible at the time the gold standard was established [95]. the platinum standard that predicts future collagen deposition must be established.

Nevertheless, the collagen reducing effect within the experimental period appears to be indispensable for a go-no-go decision in a preclinical examination. What are the requirements to be the winner in the current examination in animals? Since such power of collagen accumulation during the short period is far beyond physiological and even pathological regulation of animals, one winner of the decision is an agent that disrupts collagen production network, and another is an agent that reduces viability of HSCs, and a looser could be a blockade for a specific pathway of collagen production or an agent that indirectly reduces collagens by, for example, supply of myofibroblasts. Considering the chronic nature of human fibrosis, an agent that inhibits profibrotic process gently but steadily is an important choice, which is contradictory to the acute collagen reducing efficacy required for the decision.

### 9.2. Target Specificity of Integrin Inhibitors and Pathology Specificity of Target Integrins

Learned from the developmental history, not all, but many of the αv-inhibitors have an inherent nonspecific property with regard to binding of inhibitors to the target and engagement of the target integrin to ligands. In contrast, the integrin inhibitors already in the clinic, such as abciximab (targeting αIIbβ3 on platelets) [96], natalizumab (α4β1 on T-cells) [97,98], vedolizumab (α4β7 on T-cells) [99], and lifitegrast (αLβ2 on T-cells) [100] commonly bind specifically to their targets. Three of the inhibitors are target specific mAbs, and the other inhibitor, liftegrast, is localized in the site of pathology (dry eye) due to the nature of the ophthalmic solution. In these conditions, the target integrins are close or in the site of pathology, as a platelet itself causes the pathology and T-cells migrate in and close to the site of pathology. The characteristics from these successful inhibitors suggest that targeting pathology-specific role of integrins is achieved rather by topological factors than functional contrivance. In this point of view, integrins distinctively expressed in activated fibroblasts/HSCs, α8β1 and α11β1, are ideal therapeutic targets in fibrosis. The demarcated expression in pathological tissue must be quite favorable as clinical integrin inhibitors.

### 9.3. Combination Therapy

There are a lot of diverse pathways to fibrosis, where intra-cellular signals influence one another. A single blockade of the pathways in the network may easily be bypassed or even activate another pathway to develop fibrosis. Clinical trials employing multiple drugs in combination are underway, which may be required to reach effective therapy. This strategy could be applied within integrin inhibitors. For example, since TGFβ activation was found recently to be exerted also by α8β1 [11], shutting down all the αv integrin signals may not completely block TGFβ activation. Blocking the α8β1-mediated pathway could greatly enhance the effects of pan-αv inhibitor. Alternatively, continuous (and low dose) administration of a well-tolerated pathology-specific drug in combination with intermittent administrations of a drug with potent effects, such as pan-αv could be one combination.

One more conceivable effective combination is targeting the pathology-specific integrins together. The mAbs for α8β1 and α11β1 are both expected to inhibit myofibroblast differentiation [11,12,86] through distinct pathways, a signal mediated by α8β1 is initiated by engagement with RGD-containing proteins such as nephronectin and α11β1 by collagens. Because the expression of these integrins is induced in activated HSCs and activated HSCs with myofibroblastic phenotype appears only in fibrotic tissue, the combination is favorable also in terms of safety. Once myofibroblast differentiation is blocked, fibrotic tissue is starved for the effector cells, and little matrix proteins are newly deposited. This is, therefore, an attractive application of the combination therapy in terms of effects and safety. Of course, a combination of integrin inhibitors with drugs with a different mechanism, such as modulators for lipid metabolism, is also an expected option. Since combination therapy is commonly used in the cancer chemotherapy, the strategy appears to be adequate to apply to the highly retractable liver fibrosis, where there are no effective drugs on the market.

### 9.4. Allosteric Inhibition

A unique feature in the approved integrin inhibitors is that the target integrins are all expressed on circulating cells, i.e., platelets and leukocytes. On the contrary, target integrins of anti-fibrotic drugs are expressed in the tissue cells such as epithelial cells and fibroblasts. Importantly, the mode of actions of integrin inhibitors may not be the same by cell type expressing the target. Most circulating cell integrins are not occupied by ligands but prepared for attachment to vascular endothelial cells with the ligand-binding pocket open to interact with ligands such as VCAM-1, ICAM-1 and MadCAM-1.

On the other hand, integrins expressed in solid tissues are largely engaged with ligands in the tissue, such as fibronectin, vitronectin, tenascin-C, and nephronectin, or laminins and collagens. In this circumstance, the inhibitor for matrix receptor integrins must dissociate the ligand from integrins to block signal input [91,92], while those for circulating cell integrins just cover the ligand-binding pockets to perform their own tasks. Notably, the dissociation of ligands from cells is achieved by allosteric inhibition, which is clearly illustrated in the action of anti-α4 neutralizing mAb, natalizumab [101]. Allosteric inhibition occurs by binding not directly to the ligand-binding pocket but to the site nearby the pocket, as the antibody clashes with the ligand occupying the pocket (Figure 7). Although this mechanism of action needs to be clarified more precisely, there are 2 types of neutralizing mAb for integrins: one detaches the cell and the other does not. If this is the case, the small molecule inhibitors that occupy the ligand-binding pocket would have no effect on integrins expressed on tissue cells. Because some of the neutralizing mAbs exhibit allosteric inhibition (Figure 6), such mAbs may be a prior modality to a small molecule to shut down signals from tissue cell integrins. How an allosteric inhibitor mAb against integrin disrupts existing binding between the integrin and ligand is thoroughly described [101].

## 10. Concluding Remarks

Inhibition of integrins attenuates fibrosis in preclinical studies [8,9,10,32,33,34,39,46,47,48,56,57,102]. Integrins are not playing roles prominently in fibrogenesis but are closely related to or even regulate leading performers, matrix proteins, fibroblasts, and TGFβ, and thus serve as one of few subsets of therapeutic targets of fibrosis. Because the pro-TGFβ activation in situ is so inspiring for the design concept of anti-fibrotic drugs, αv-integrins have been a central target for liver fibrosis, leaving many other integrins behind. However, being no αv inhibitor drugs in the clinic, more pathology-specific drugs and targets have to be explored. There are stocks of integrins that fulfill the pathology-specific property as targets, α8β1 and α11β1. Either the cell type-specific expression in fibroblasts or overwhelming upregulation in activated HSCs were not found in any other members of the integrin family. Both integrins in fact exhibit profibrotic properties such as myofibroblast differentiation, and α8β1 activates TGFβ and α11β1 may do so [103]. The neutralizing mAb for each integrin readily allows further validation in vivo and in vitro. Besides the inhibitory effects, the target’s “pathology specific” induction in activated HSCs must be a distinct advantage to eliminate off-target adverse effects. Whether the mAbs enhance anti-fibrotic effects of αv-inhibitors may be an interesting option. 

What is at all the role of integrins that are expressed specifically on activated HSCs? (Figure 8) The answer must be within the biological missions of HSCs.

## 11. Patents

PCT/JP2010/068374 and PCT/JP2013/059368 are patents for the anti-α8β1 mAb, and PCT/JP2019/008202 is for the anti-α11β1 mAb.

## Figures and Tables

**Figure 1 ijms-22-12794-f001:**
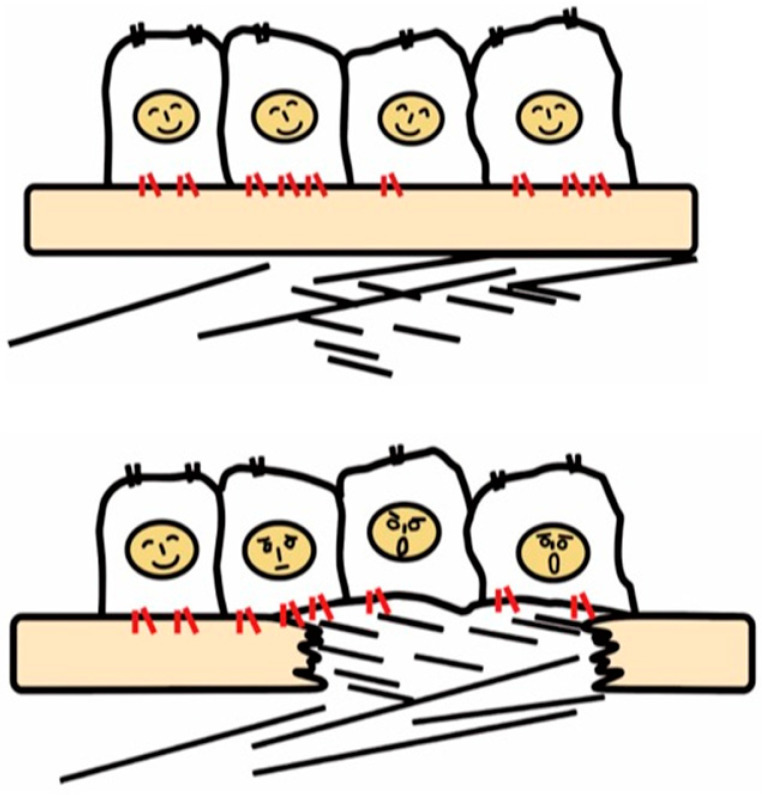
**Recognition of tissue injury by epithelial cells via integrin receptors.** In the healthy tissue (upper panel), epithelial cells know their peaceful circumstance recognizing components of the basement membrane such as laminin via integrins (red). Upon tissue injury, cells recognize contact with unusual matrix proteins such as collagen type I and fibronectin, which are normally in sub epithelium, and notice the emergent condition.

**Figure 2 ijms-22-12794-f002:**
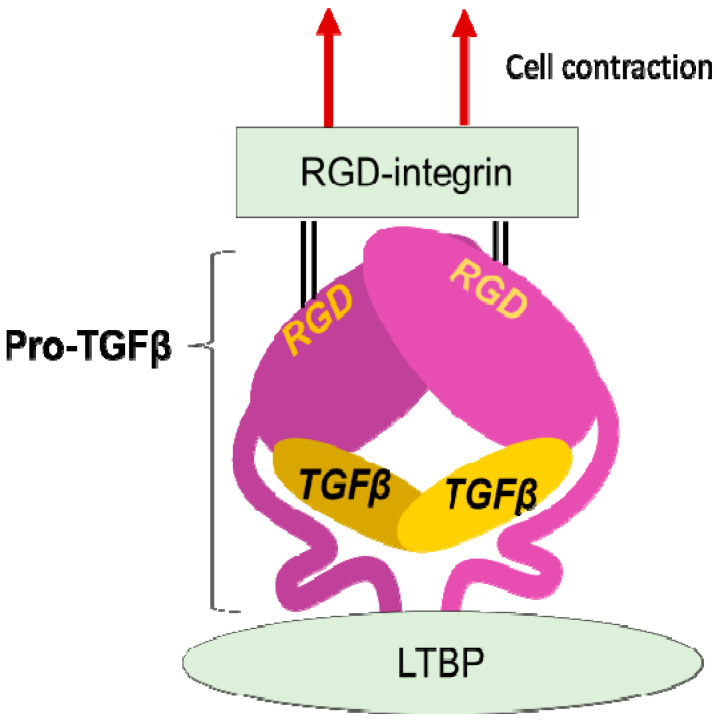
**Integrin mediated TGFβ activation.** TGFβ is stored in the extracellular milieu anchoring to LTBP that is fixed to matrix proteins, as a pro-protein, also termed as LAP. The pro-TGFβ protein forms homodimer holding TGFβ by the pro-domains. There is an RGD sequence in the pro-domain. To release TGFβ, RGD-recognizing integrins binds to the RGD sequences and pro-domains are removed from the matured TGFβ by cellular tensile force initiated by contraction of the cell expressing the integrins.

**Figure 3 ijms-22-12794-f003:**
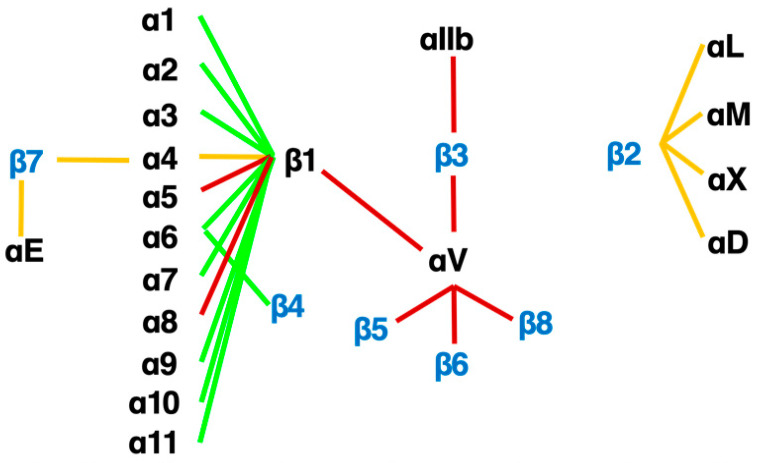
**The integrin family.** Twenty-four heterodimers and the combinations of 18 α and 8 β subunits are indicated. Eight RGD-recognizing integrins are paired with red lines. Orange lines indicate leukocyte integrins. An alternative name for αIIbβ3 is GPIIb/IIIa, and for αLβ2 are LFA1 and CD11a/18.

**Figure 4 ijms-22-12794-f004:**
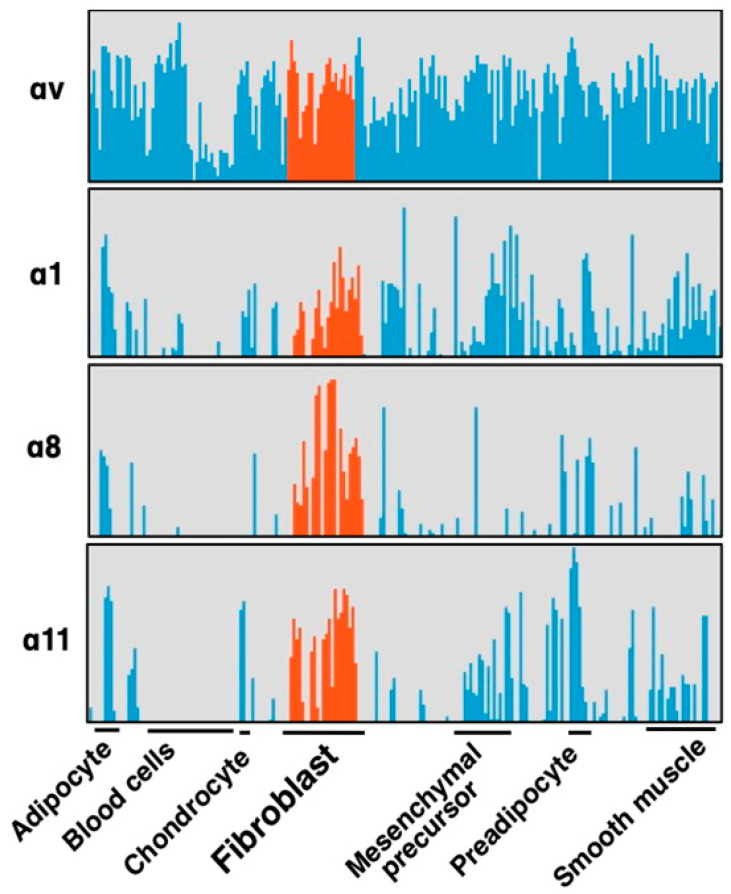
**Selective expression of integrin α subunits in fibroblasts.** Four α subunits, α1, α8, α11 and αv are upregulated by fibrotic stimulation in HSCs (Figure 5). Expression of the subunits in primary cultures of various cell types are compared focusing on fibroblasts (red). These 4 subunits are expressed in primary cultured fibroblasts from various tissues, and α8 and α11 are in the fibroblast selective manner, while αv subunits are ubiquitously expressed across cell types. α1 is expressed in fibroblasts but also in non-mesenchymal cells such as leukocytes and endothelial cells. α8 and α11 are principally expressed only in mesenchymal cells and α8 is the most specific subunit to fibroblast compared with α11.

**Figure 5 ijms-22-12794-f005:**
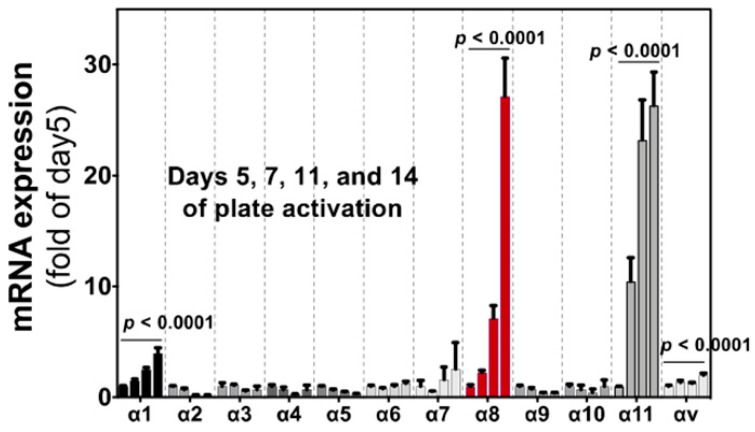
**Induction of integrin α subunit in HSCs by culture activation.** Bars indicate relative expression of day 5 for α subunits indicated by qPCR. All α subunits that are expressed in tissue cells (excluding cells in the circulation) are evaluated. Each bar represents mean ± SE, and statistical significances were calculated by ANOVA. (Adopted from Ref. [11]).

**Figure 6 ijms-22-12794-f006:**
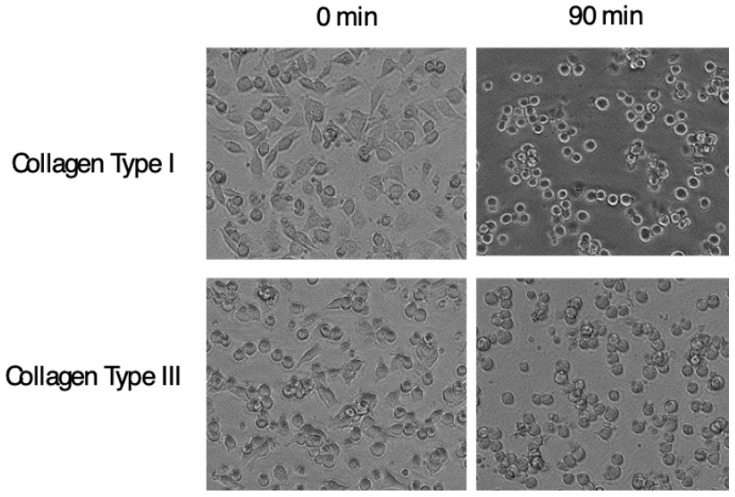
**Cell detaching effect of anti-α11 neutralizing mAb**. C2C12 cells transfected with *Itga11* cDNA were plated on plates coated with indicated collagen. After cells adhered, the anti-α11 mAb, YW33, was added into the culture. Pictures in the left column show cells adhered to the plate, and in the right show cells 90 min after the mAb input. Changes in cell shapes into rounded form indicate weakened attachment of cells.

**Figure 7 ijms-22-12794-f007:**
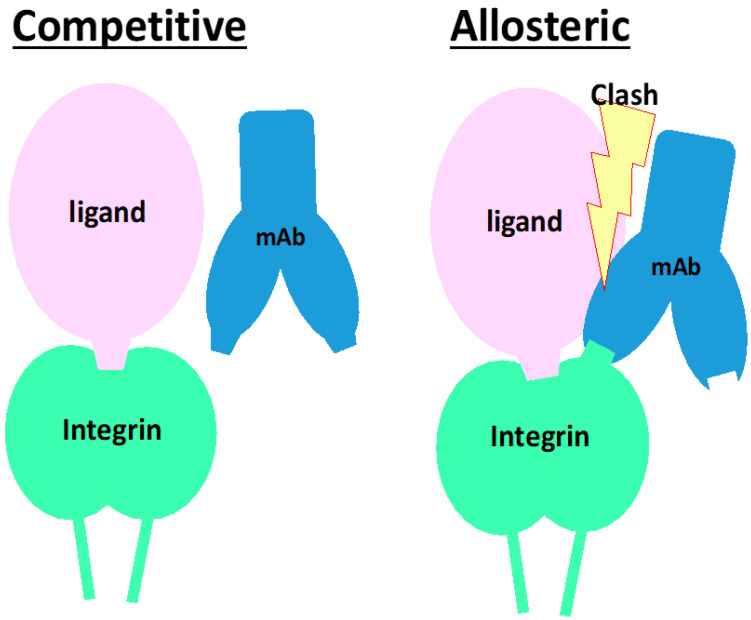
**Schematic of mode of actions for neutralizing mAb with competitive or allosteric inhibition.** In both left and right panels, a ligand binds and occupies the ligand-binding pocket of an integrin, encompassing the α and β subunits of integrin. Left panel: The mAb competitive inhibitor binds to in and periphery of the ligand-binding pocket. When a ligand is already binding to the pocket, the mAb does not bind or even inaccessible to the epitope. The epitope is completely covered with the ligand. Right panel: The mAb binds to integrin at the epitope that is not as close as the competitive inhibitory mAb but localizes closely but around the ligand binding pocket. The mAb is allowed to bind to the integrin while clashing with the ligand.

**Figure 8 ijms-22-12794-f008:**
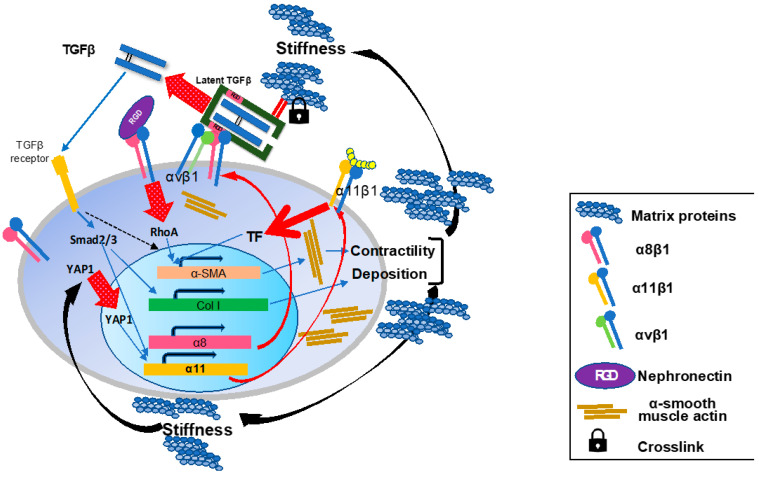
**Schematic summary of the roles of the integrins on activated hepatic stellate cells in liver fibrosis.** During the fibrogenesis, α8 and α11 subunits are induced on activated HSCs/myofibroblasts. In the fibrotic milieu of the tissue, increased matrix rigidity induces intranuclear translocation of YAP1 and initiates transcription of *Itga11* gene. Induced a8b1 and α11β1 both promote a-SMA expression, actin fiber formation, and cellular contractility upon ligand-engagement with such as nephronectin and collagen type I, respectively. Once HSCs acquire the myofibroblast phenotype, the cells contribute more to TGFb activation on their surface through interactions of integrins including a8b1 and αvβ1 with pro-TGFb in the surrounding matrix proteins. Released matured TGFb binds to its receptor and initiates the Smad signaling cascade. The signal promotes production of matrix proteins containing collagen species from myofibroblasts and α11 expression. Crosstalk between TGFβ-initiated and a8b1- and α11β1-mediated signals cooperatively enhance a-SMA expression. These effects by a8b1 and α11β1 on HSCs consequently render myofibroblasts highly contractile and productive for collagens and other matrix proteins, which reinforces tissue stiffness to maintain and enhance a11 expression.

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
