# Peer review of "New Therapeutic Targets for Hepatic Fibrosis in the Integrin Family, α8β1 and α11β1, Induced Specifically on Activated Stellate Cells"

_ijms, 2021, doi:10.3390/ijms222312794_

Round 1

Reviewer 1 Report

This is a very interesting review on the role of integrins as potential targets in the treatment of fibrosis mainly in NAFLD/NASH associated fibrotic process.There is a thorough presentation of the subject both on the basic aspects and the potential clinical applications. It is most useful for all those engaged in either basic research of fibrosis or the clinical handling of those patients. The individual parts are covered in depth and the information presented is up to date. Moreover the figures presented are very usuful and informative. However there are two points that require attention:

1) Line 35: Liver fibrosis is not synonymous to cirrhosis as in cirrhosis the presence of regenerative nodules is mandatory in contrast to liver fibrosis. Moreover liver fibrosis is not always intractable as the authors state. NAFLD/NASH is not the only etiology of fibrosis. For example fibrosis is reversible in viral hepatitis after elimination of the virus and the same is true in some cases of alcoholic fibrosis after alcohol withdrawal. These points should be clarified.

2) The english language requires extensive polishing and in some cases re-writing in plain english is required as they are incromprehensible at times.

Author Response

Thank you very much for your favorable and valuable comment to the manuscript.

We appreciate the correction of our mistake in the "cirrhosis" usage at the first line of the text. We revised as you suggested. With respect to the language issue, a native speaker physician scientist in the US checked for the language of the manuscript for us. 

Reviewer 2 Report

The authors have reviewed the literature about αv-integrins and how they can be targeted for treating fibrosis. The authors have discussed specifically various aspects of α8β1 and α11β1 integrins, including the expression pattern on activated stellate cells, potential to activate TGFβ pathway, distinct ligands, and role in fibrosis. Moreover, the authors have also proposed an interesting possibility of combination therapy using neutralizing monoclonal antibodies for α8β1 and α11β1 for the effective treatment of fibrotic diseases. Overall, I recommend the publication of this review.

Author Response

Thank you for your favorable comments.  We have checked the language and checked the spell.

Reviewer 3 Report

The Authors present a paper entitled “New therapeutic targets for hepatic fibrosis in the integrin family, α8β1 2 and α11β1, induced specifically on activated stellate cells”. The Authors review the  efforts devoted to developing drugs targeting integrins as a central target of fibrosis. A special focus has been dedicated to the revision of monoclonal antibodies directed against α8β1 and α11β1 in preclinical examination. The topic is of extreme interest as it covers different diseases,. The problem is well presented and the background largely described. The language is appropriate with only minor spell typos. I would recommend to accept the review in the present form, nevertheless to allow a fully understanding of the topic, I would suggest the Authors to insert a table to summarize the different types of integrin with their function.

Author Response

Thank you for your favorable comments. We have checked the spell again and revised a few spells.